# Alarm-assisted urotherapy for daytime urinary incontinence in children: A meta-analysis

Liesbeth L. de Wall [1]*, Antje J. Nieuwhof-Leppink[2], Renske Schappin[3]

1 Department of Urology, Radboud University Medical Centre Amalia Children's Hospital, Nijmegen, The Netherlands, 2 Department of Medical Psychology and Social Work, Wilhelmina Children's Hospital Utrecht, Utrecht, The Netherlands, 3 Department of Surgery, Wilhelmina Children's Hospital Utrecht, Utrecht, The Netherlands

☯ These authors contributed equally to this work.
* liesbeth.dewall@radboudumc.nl

## Abstract

**Data Availability Statement:** Data are available on the DANS easy data repository (DOI: 10.17026/dans-zt8-2t4a).

### Objectives

Wearable alarm systems are frequently used tools added to urotherapy for children with both daytime and nighttime urinary incontinence. For functional daytime incontinence (DUI) specifically, the effect of alarm interventions has not been systematically reviewed. This study systematically evaluates, summarizes, reviews, and analyzes existing evidence about the effect of wearable alarm systems in urotherapy for children with functional DUI.

### Study design

We completed a comprehensive literature search in August 2022 using MEDLINE/PUBMED, EMBASE, PsycINFO, Cochrane Library, Web of Science, Google Scholar, conference abstracts, and citation tracking. Clinical controlled trials at controlled-trials.com and clinicaltrials.gov were consulted, as was the National health Service Center For Reviews And Dissemination. Eligible studies including the use of noninvasive wearable alarm systems as (part of) treatment for functional DUI in children were included. The main outcome was continence after treatment. Three independent reviewers extracted data. Risk of bias was assessed using Cochrane and National Heart, Lung and Blood Institute quality assessment tools.

### Results

A total of 10 studies out of 1,382 records were included. Meta-analysis revealed a nonsignificant risk ratio of 1.4 (95% CI: 0.8–2.6) for the use of alarm systems. Urotherapy with alarm systems resulted in a 48% (95% CI: 33–62%) continence rate after treatment.

**Funding:** The authors received no specific funding for this work.

**Competing interests:** The authors have declared that no competing interests exist.

**Abbreviations:** CBT, Cognitive behavioral therapy; CI, Confidence interval; DUI, Daytime urinary incontinence; GRADE, The Grading of Recommendation Assessment, Development, and Evaluation; ICCS, International Children's Continence Society; LUTD, Lower urinary tract dysfunction; MH, Mantel-Haenszel; NHLBI, National Heart, Lung and Blood Institute; PRISMA, Preferred Reporting of Items for Systematic Reviews and Meta-Analysis; RCT, Randomized controlled trial; Rob2, Version 2 of the Cochrane risk-of-bias tool for randomized trials; SD, Standard deviation; UTI, Urinary tract infection.

## Conclusions

Alarm systems might be helpful as part of urotherapy for functional DUI in select cases. Adherence is problematic, and the optimal duration of the use of alarm systems is to be determined. Overall, the risk of bias was high in all studies.

## Introduction

Functional daytime urinary incontinence (DUI) is a common problem in school-aged children and a frequent condition in pediatric health care [1,2]. In 7-year-old children, the prevalence varies between 3.9 and 9.0% with approximately 0.5 to 0.8% suffering from severe DUI [1,3]. Children may experience psychological distress, decreased self-esteem, and decreased self-confidence due to DUI, with increased risk of social isolation [4,5]. Parents are often concerned about their child's wellbeing. Furthermore, persistent urinary incontinence is associated with an increased risk of lower urinary tract dysfunction (LUTD) in adolescence and even adulthood [6,7].

According to the International Children's Continence Society (ICCS), urotherapy is the first-line treatment for functional LUTD in children [8,9]. It is a conservative-based therapy with rehabilitation of the lower urinary tract as the main goal. Urotherapy includes providing information and instructions, behavioral modification, life-style advice, and registration and monitoring of drinking and voiding habits [10]. Approximately 56% of children with DUI achieve continence within a year by means of urotherapy compared to a spontaneous cure rate of 15% per year [11]. Other treatment options include cognitive behavioral therapy (CBT), biofeedback training, pelvic floor muscle training, neuromodulation, and medication [9].

Wearable alarm systems like timer watches and alarm pants are frequently used tools in urotherapy [9]. They are intended to teach children to recognize the sensation of a full bladder and to stimulate appropriate toilet behavior. The timer watch gives a signal at set intervals at which the child should void, increasing the child's regular toilet visits and aiming to prevent wetting accidents [12]. The alarm pants give a notification when a wetting accident has already occurred. This provides feedback that the child missed a full bladder signal. The child learns to recognize which bladder sensation precedes a wetting accident, indicating when to void.

From a historical perspective, alarm pants were first successfully introduced in the context of children wetting their beds at night [13,14]. Based on the positive results obtained in this condition, physicians started to use alarm systems for DUI as well. However, research on the effectiveness of alarm systems in children with functional DUI is scarce and heterogeneous. Published data differ in the type of alarm system used (timer watch or alarm pants), condition causing DUI, population, outcome parameter(s), and treatment duration [15–17].

The aim of this systematic review is to study the following research question: "What is the additional effect of wearable alarm systems used in urotherapy for functional DUI in otherwise healthy school-aged children compared to urotherapy alone?" Additionally, we investigate adherence to alarm interventions, difference in effectiveness between types of alarm, relapse during follow-up, and differences between treatment duration on outcome.

## Methods

This systematic review is reported according to the Preferred Reporting of Items for Systematic Reviews and Meta-Analysis (PRISMA) protocol modified for the pediatric population [18,19].

Our protocol was registered with Open Science Framework, registration number 10.17605/OSF.IO/BG8H9.

## Search strategy

We conducted a comprehensive literature search in April 2021 and repeated it in August 2022 using the following databases: MEDLINE/PUBMED, EMBASE, PsycINFO, Google Scholar, Cochrane Library, Web of Science, conference abstracts of the European Society of Pediatric Urology, and citation tracking. We looked for clinical controlled trials at controlled-trials.com, clinicaltrials.gov, and the National Health Service Center for Reviews and Dissemination. Reference lists of all included studies were scanned. Unpublished studies were sought by locating conference abstracts and contacting authors. We used Boolean logic to incorporate synonyms for concepts that define the use of alarm systems in the treatment of pediatric functional DUI. We combined three filters: 1) DUI, 2) urotherapy with the use of noninvasive alarm systems and synonyms for urotherapy (voiding school, bladder training, or bladder rehabilitation), and 3) children. For the latter, we used Leclercq et al.'s pediatric search filter [19]. The full electronic search strategy can be found as supporting information "S1 Appendix" Controlled vocabulary (EMTREE in Embase and MeSH in PubMed) and keywords were used when possible.

## Selection procedure

All articles (published or unpublished) that use or mention the use of noninvasive wearable alarm systems as (part of) treatment for functional DUI were included for further analysis. Other inclusion criteria were 1) a report of at least one outcome measurement; 2) age between 4 and 18 years; 3) written in English, German, or Spanish; and 4) available as a full text. Prior or concomitant treatment for LUTD (e.g., pelvic floor training, medication, treatment for constipation, neuromodulation, cystoscopy to exclude anatomical obstruction, treatment for urinary tract infections) was deemed acceptable because such treatment is typical in this population. Studies published in the last 40 years were included for further analysis. The overall search result of each database was transferred into a folder in Rayyan® where duplicates generated by journals indexed in more than one database were automatically deleted [20].

## Exclusion criteria

We excluded monosymptomatic enuresis because monosymptomatic enuresis and DUI are considered two different conditions each having a different etiology and treatment according the ICCS [9,21]. Other exclusion criteria were 1) DUI due to neurological conditions or significant urogenital anatomical abnormalities, 2) concomitant intravesical botulinum toxin A treatment, 3) non-potty-trained children (this is not regarded as functional DUI) [22], and 4) children with intellectual and/or developmental disabilities. The latter were excluded because training of these children involves a different approach and different goals [17].

## Data extraction and quality assessment

All authors independently screened all abstracts and titles and completed full text analyses. Disagreements were resolved through discussion of the topic and consensus. Extracted data from the selected studies were organized in an Excel spreadsheet following the checklist Consolidation of the Standards of Report Trial-CONSORT [23]. The Cochrane collaboration risk-of-bias RoB-2 tool was used to assess the risk of bias of the randomized controlled trials (RCTs) at the outcome level. Because this systematic review also includes observational studies,

internal bias risk was quantified according the National Heart, Lung and Blood Institute (NHLBI) quality assessment tool for before–after studies without a control group and the Joanna Briggs Institute Critical Appraisal tool for case reports, both at the outcome level [24–28]. The Grading of Recommendation Assessment, Development, and Evaluation (GRADE) approach was used to assess the quality of evidence of all included studies [29]. All authors independently assessed the articles and resolved any differences by discussion.

## Primary outcomes

The primary outcome parameter is continence expressed as the percentage of children who were continent after treatment. The operationalization of continence is according to the ICCS standards [8,9]. Continence is measured as an improvement of symptom frequency from baseline to the end of treatment. Symptoms are urgency, flow pattern, voiding frequency, and wetting accidents. Usually, only wetting accidents are used to assess improvement. The international standardization documents published by the ICCS classify improvement into no response (<50% reduction), partial response (50 to 99% reduction), and complete response (100% reduction). The former term 'response' (>90% reduction) is no longer used and is now reclassified as partial response [9]. Before the ICCS guidelines were established (2008), continence was usually measured as the reduction in wetting accidents from baseline to the end of treatment.

## Secondary outcomes

Secondary outcomes were relapse (the percentage of children in whom DUI reoccurred after they had initially achieved complete response), type of alarm, duration of alarm treatment, and adherence (expressed as a percentage). There is no standardized measure of adherence in this field. Therefore, in each study, the definition mentioned by the authors was used.

## Data synthesis and statistical analysis

For the meta-analysis, effect estimates were pooled using the Mantel-Haenszel (MH) method, and a random effect model was employed for the included RCTs to study whether urotherapy with the use of an alarm system was more effective than urotherapy without the use of an alarm system. For the observational studies and the intervention groups of the RCT studies, we used a random effect model with inverse variance to study the overall proportion of continence achieved with the use of any type of alarm. We applied an intention-to-treat analysis with drop-out cases considered to remain incontinent after treatment. The software used to analyze the data was Comprehensive Meta-Analysis V3 [30].

Outcome data were expressed as an MH risk ratio or event rate with 95% confidence intervals (CIs). We used a two-sided level of significance of $P < 0.05$. The statistical heterogeneity among studies was evaluated using $Q$ and $I^2$ statistics. A value greater than 50% was suggestive of substantial study heterogeneity.

We conducted subgroup analyses on the type of alarm system used (pants alarm versus timer watch) in a random effect model with pooled random effects variance. To investigate whether relapse, treatment duration, or adherence were relevant predictors that could account for the variance component $Tau^2$ in the overall event rate, we performed a meta-regression analysis assuming a random effect model using a $Z$ distribution. The regression coefficient $R^2$ statistic was used as a measure of the proportion of the observed variance that was not due to sampling error. We examined publication bias through visual funnel plot inspection, trim-and-fill procedure, and Egger's test.

## Results

The overall literature search is shown in Fig 1. A total of 2,549 records were identified from database searches and three records by manual searching of abstracts. After removal of duplicates, 1,382 studies remained, and after screening of abstracts and titles and reading of full-text articles, 10 studies were eligible for inclusion in the qualitative analysis. A total of nine studies were eligible for quantitative analysis because one study, being a case report, was excluded. For two studies [31,32] the long term outcome were published during follow-up and data were included [33–35].

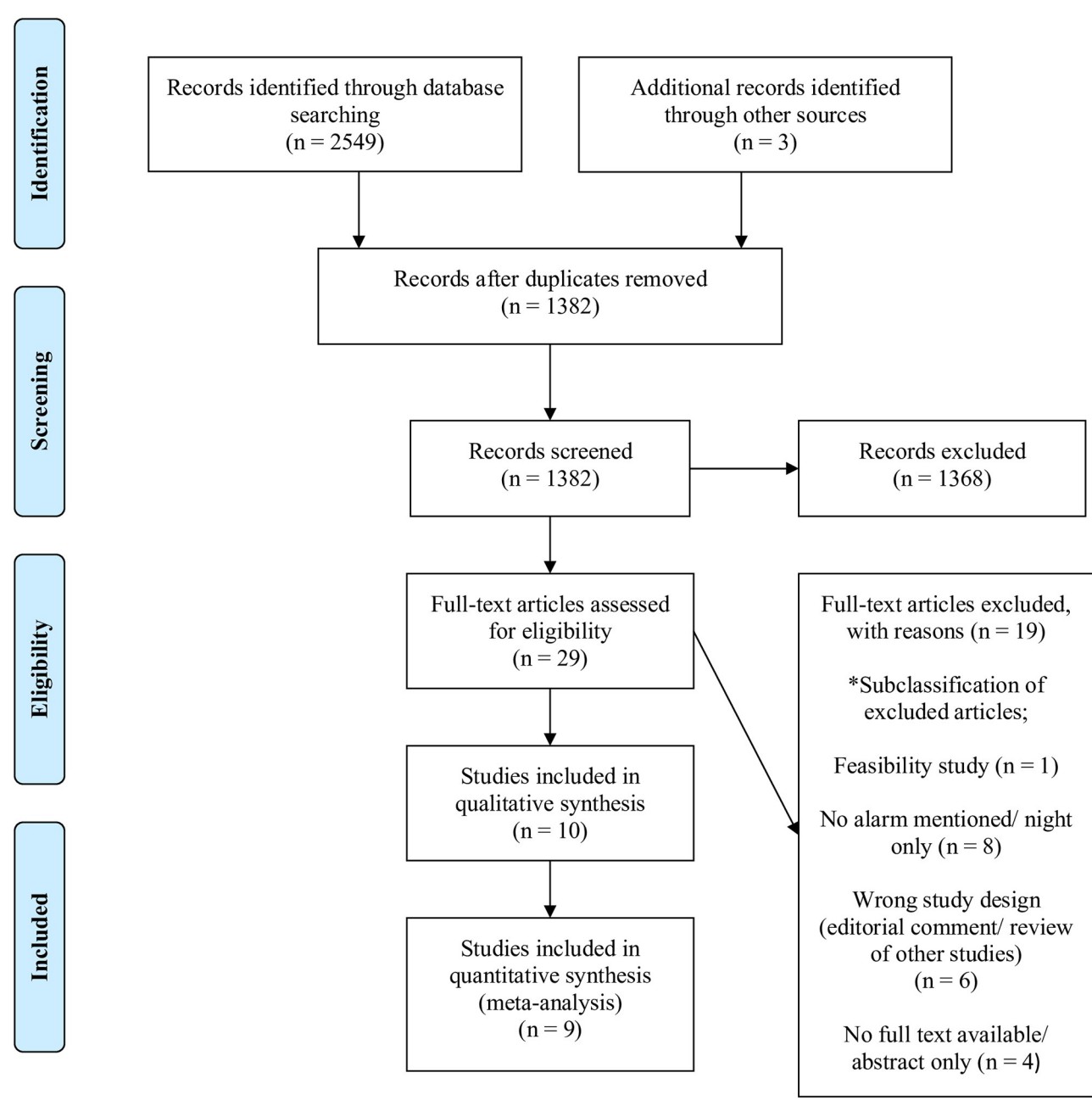

**Fig 1. Flow diagram of the selection of eligible studies according to PRISMA guidelines.**

## Study characteristics

Study characteristics, design, and outcomes are shown in Table 1. We identified four studies with a control group, three of which were RCTs [12,31–33]. Of the other six studies, four were retrospective chart studies [16,34–36]. There was one case series of four subjects and one case report [37,38]. Most studies included children with functional DUI refractory to previous treatment, and two studies included therapy-naïve children [33,37]. Sample sizes varied from 1 to 226 children. In three of the ten studies, a timer watch was used as alarm device; the remainder of the studies used a pants alarm [12,33,34]. Five studies used percentage response (complete, partial, no response) as their primary outcome measurement as recommended by the ICCS [8,9]. The remainder used the percentage of children that became continent with different definitions of continence used [31,32,35,37,38]. Vijverberg et al. included micturition frequency, urge complaints, and flow pattern in their outcome besides the number of wetting accidents before and after treatment and reclassified the outcomes of the different parameters into having a good, average, or bad response [35]. Adherence was described in four of the nine studies [12,31,33,37].

## Methodological quality of the included studies

The overall risk of bias was high in all studies (Fig 2A and 2B). A considerable number of patients were nonadherent and therefore did not receive the intervention as intended by the authors. In the RCT by Halliday et al., both patients and professionals were blinded to group allocation, while there was no blinding in the RCT by Hagstroem et al. and selective blinding of investigators/data analysts only in the RCT by Caldwell et al. [12,31,33]. In all observational studies, bias due to confounding was high. The effect of urotherapy was studied with alarm therapy included as part of a broader protocol and not specifically the singular effect of alarm treatment. Subsequently, factors other than alarm treatment might be responsible for the measured outcome. In most nonrandomized studies, there was a high risk of bias in the selection of subjects because these were clinical studies in which children received alarm treatment if the professional deemed it the best treatment option for the specific child. In addition, inclusion and exclusion criteria were often not comprehensive or replicable. For all included studies, continence is an inherently subjective outcome measure because it is generally based on self-reported measurements like voiding diaries.

## Primary study outcome

**Continence rate.** The effect of wearable alarm systems on continence, defined as a complete response (100% dry), was determined in two ways. A meta-analytic random model of all included RCTs revealed an RR of 1.2 (95% CI: 0.8–1.9) for the study by Halliday et al. (comparing a noncontingent alarm with a contingent alarm), an RR of 17.7 (95% CI: 1.0–291.8) for the study by Hagstroem et al. (comparing urotherapy with or without a timer watch), and an RR of 1.3 (95% CI 0.8–2.2) for the study by Caldwell et al. (comparing a timer watch with a standard watch without an alarm) [12,31,33]. The pooled RR of the RCTs was 1.4 (95% CI: 0.8–2.6) with a Z-value of 1.13 (P = 0.257). Training with a contingent alarm or timer watch did not significantly increase the outcome of being continent after training (Fig 3A). Because the control group of the RCT by Halliday et al. (noncontingent alarm) is comparable to the intervention group in the other two RCTs (timer watch), the moderate to high residual heterogeneity of effect sizes ($I^2$ = 54%) is partially explained. A random effect model (with a combined effect if multiple measurements in time were available) was used to study the proportion of continent children after treatment with alarm systems. Analysis revealed a 48% continence rate (95% CI: 33–62%; Fig 3B). Residual heterogeneity of effect sizes was high at $I^2$ = 87%.

**Table 1. Study characteristics of all studies included in this systematic review.**

| Study details (author/ year/state) | Study design | Inclusion criteria | Number of patients | Mean age in years (SD or range) | Type of alarm | Treatment duration (days)** | Non adherence (%) | Definition of continence | Outcome directly after treatment | Long-term outcome |
|---|---|---|---|---|---|---|---|---|---|---|
| Boelens et al. (2003) [37] Netherlands | Case series | DUI (at least 4 accidents/ week) | 4 | 6.5 (2.1) | Pants alarm | Mean 75 (SD 27) | 75% | 14 consecutive days without accidents | 25% dry | - |
| Hagstroem et al. (2010) [12] Denmark | RCT | Refractory OAB with DUI (at least 1 accident/ week) | *Intervention* 30 | 7.6 (1.7) | *Intervention* Timer watch | Protocol: 84 | *Intervention* 33% | ICCS | *Intervention* 30% CR 30% PR | *Intervention group only* After median 9.5 weeks: 59% CR 11% PR After median 7 months: 67% CR 0% PR |
| | | | *Control* 28 | | *Control* Timed voiding every 2 hours | | *Control* 67% | | *Control* 0% CR 18% PR | |
| Hagstroem et al. (2008) [34] Denmark | Retrospective study | Refractory DUI (median 7.8 accidents/ week) | 60 | 7.7 (1.6) | Timer watch | Median 114 (25th–75th Quartiles 64–146) | - | ICCS | 70% CR | - |
| Halliday et al. (1987) [31] UK | RCT | Refractory DUI (mean 9.5 accidents/ week) | *Intervention* 22 *Control* 22 | 8.5 (2.1) | *Intervention* Pants alarm *Control* Noncontingent alarm every 2 hours | Protocol: 84 Intervention terminated when child was continent | 10%*** | 6 consecutive weeks without daytime wetting | *Intervention* 73% dry *Control* 59% dry | - |
| Van Laecke et al. (2006) [16] Belgium | Retrospective study | Refractory OAB with DUI (1 to 5 accidents/ day) | 15 out of 63 trained with alarm (no concomitant therapy) | 8 (range 5 to 14) | Pants alarm | Protocol: 14 | - | ICCS | 47% CR 13% PR | 6 weeks after end of training: 40% CR 20% PR 1 year after end of training: 27% CR 33% PR |
| Vijverberg et al. (1997) [35] *Follow-up data* Vijverberg et al. (2011) Netherlands | Retrospective study | Refractory DUI and recurrent UTI | 95 | 9.9 (2.4) | Pants alarm | Protocol: 10 | - | Good: ≤ 1 accident/ week, normal flow and micturition frequency, no urge. Average: < 2 accidents/ week, normal flow, improved micturition frequency and urge. | - | 6 months after training: 68% good 13% average After mean 18 years: 84% good 11% average |
| Meijer et al. (2015) [36] Netherlands | Retrospective study | Refractory OAB | 70 | 9.3 (1.5) | Pants alarm | Protocol: 10 | - | ICCS | - | 6 months after training: 43% CR 32% PR 2 years after training: 59% CR 41% PR |

*(Continued)*

**Table 1.** (Continued)

| Study details (author/year/state) | Study design | Inclusion criteria | Number of patients | Mean age in years (SD or range) | Type of alarm | Treatment duration (days)** | Non adherence (%) | Definition of continence | Outcome directly after treatment | Long-term outcome |
|---|---|---|---|---|---|---|---|---|---|---|
| Hoebeke et al. (2011) [32] *Follow up data* Van den Broeck et al. (2015) [34] Dossche et al. (2020) Belgium | Prospective case-control study | Refractory DUI* (mean 4.2 accidents/week) | *Intervention* 30 *Control* 11 | 9.0 (1.9) | *Intervention* Pants alarm *Control* Waiting list (no active therapy) | Protocol: 10 (2 x 5 days with 2-week interval) | - | No time period for being 'dry' is mentioned | - | 6 months after training: *Intervention* 53% dry *Control* 6% dry 2 years after training: *Intervention only* 60% dry |
| Friman and Vollmer (1995) [38] USA | Case report | Refractory DUI (at least 5 consecutive weeks) | 1 | 15 | Pants alarm | Protocol: 70 (2 x 35 days with 25-day interval) | - | 2 dry sanitary pads per day | 8/10 dry days | 3 and 6 months after treatment: dry |
| Caldwell et al. (2022) [33] Australia | RCT | DUI (at least twice per week for at least 2 weeks) | *Intervention* 116 *Control* 110 | 8.0 (2.1) | *Intervention* Timer watch Control Watch without alarm | Protocol: 84 | Intervention 60% Control 90% | ICCS | Intervention 22% CR Control 17% CR | 6 months after training: *Intervention* 13% CR *Control* 11% CR |

DUI = daytime urinary incontinence, OAB = overactive bladder, RCT = randomized controlled trial, UTI = urine tract infection, ICCS = International Children's Continence Society, SD = standard deviation CR = complete response (100% dry according to ICCS criteria), PR = partial response (50–99% improvement of symptoms according to ICCS criteria).

*Only children included with DUI or combined DUI/enuresis.

**Duration of treatment either according to protocol mentioned in the study or if no protocol was mentioned, median or mean days of alarm worn was used.

***No information is provided in the article about subclassification into either contingent or noncontingent alarm. (-) = no information available.

Subsequent sensitivity analyses, which removed the study by Boelens et al. because the relatively low continence rate in this study might potentially influence summary effect, revealed a summary effect comparable to the main analyses, namely a 48% continence rate (95% CI: 35–64%).

## Secondary study outcomes

**Type of alarm.** Continence rates were comparable between studies using a pants alarm or timer watch. A subgroup analysis revealed 49% continence (95% CI: 32–67%) after treatment if a pants alarm was used versus 45% continence (95% CI: 23–68%) when a timer watch was used, $Q_{between} = 0.10$, $df = 1$, $P = 0.751$, $Tau^2 = 0.63$.

**Relapse.** Relapse after complete response was mentioned in six studies and ranged between 0 and 43% [12,16,31–33,36]. Relapse was not stratified according to treatment group in the study by Halliday et al. Of all the successfully treated patients with either a contingent or noncontingent alarm, 2 of 29 children relapsed within 6 months of follow-up (7%) [31]. In Hagstroem et al.'s alarm group, none of the children with a complete response after training relapsed at a median of 7 months after treatment [12]. Van Laecke et al. report a relapse in three children (one after 6 weeks and another two after 1 year; 3/15 = 43%) [16]. Of all the

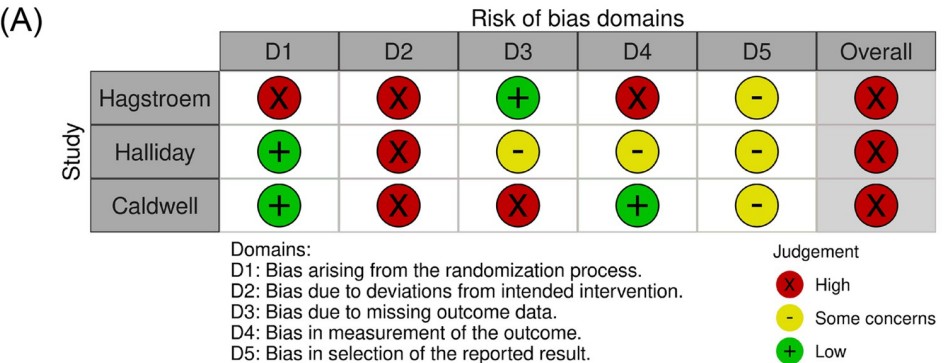

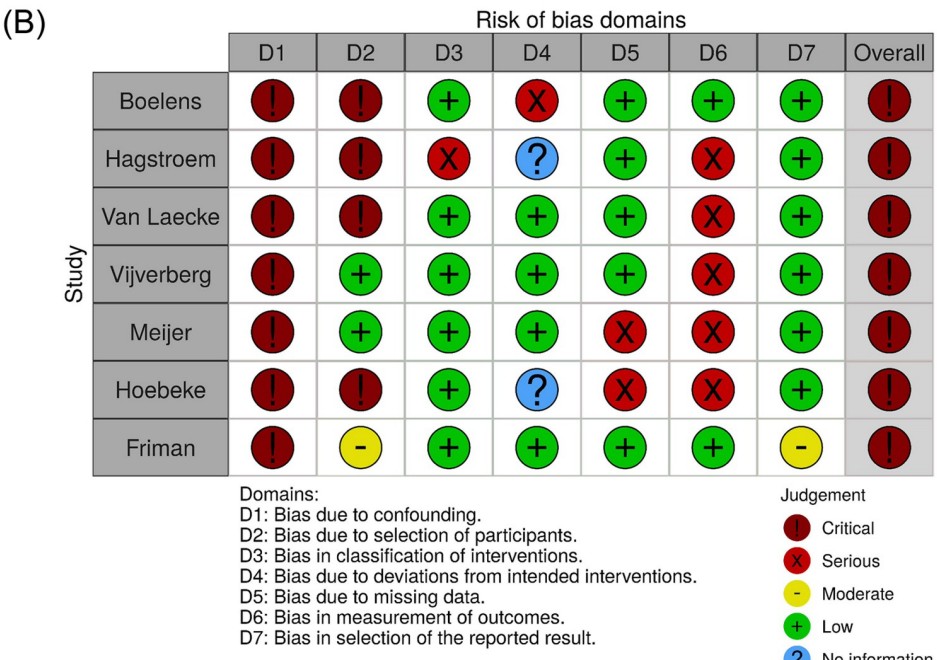

**Fig 2.** A. Risk of bias graph according to the Cochrane collaboration risk-of-bias RoB-2 tool for all included RCTs at an outcome level. B. Risk of bias graph according the National Heart, Lung and Blood Institute quality assessment tool for nonrandomized observational studies.

children who were continent after a 10-day inpatient training program, 11 children (36%) relapsed within a period of 6 months in the study by Meijer et al. [36]. In the inpatient training program by Hoebeke et al., 5 of the 16 patients who had DUI and became dry after training relapsed within 6 months of follow-up (31%) [32]. In the study by Caldwell et al., 42% (timer watch) and 33% (control) relapsed 6 months after treatment [33].

**Treatment duration.** The duration of the alarm intervention differed among studies, with some using a set period and others using the alarm until continence was achieved (see Table 1). Hagstroem et al. reported that a third of the children kept using the alarm after the official study period of 12 weeks [12]. A metaregression was conducted to examine the influence of treatment duration as a predictor of the variation across studies in continence rates. The overall proportion of variance explained by treatment duration was 0% ($R^2 = 0.00$, 95%

(A)

| Study name | Statistics for each study | | | | | MH risk ratio and 95% CI | |
|---|---|---|---|---|---|---|---|
| | MH Risk Ratio | Lower limit | Upper limit | Z-Value | P-Value | | Weight |
| Halliday et al. (1987) | 1.23 | 0.80 | 1.90 | 0.94 | 0.35 | | 50.92 |
| Hagstroem et al. (2010) | 17.7 | 1.08 | 291.82 | 2.02 | 0.04 | | 4.27 |
| Caldwell et al. (2022) | 1.30 | 0.76 | 2.21 | 0.96 | 0.34 | | 44.81 |
| | 1.41 | 0.78 | 2.57 | 1.13 | 0.26 | | |

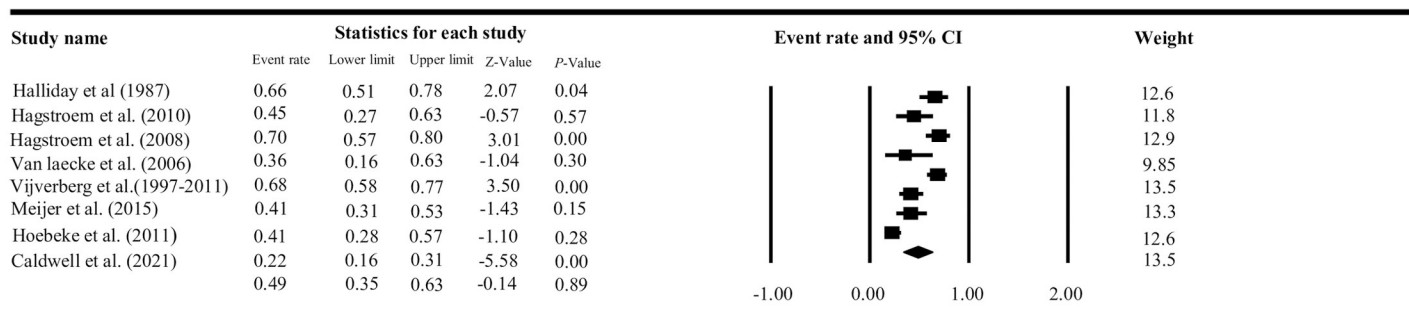

(B)

| Study name | Statistics for each study | | | | | Event rate and 95% CI | Weight |
|---|---|---|---|---|---|---|---|
| | Event rate | Lower limit | Upper limit | Z-Value | P-Value | | |
| Halliday et al (1987) | 0.66 | 0.51 | 0.78 | 2.07 | 0.04 | | 12.6 |
| Hagstroem et al. (2010) | 0.45 | 0.27 | 0.63 | -0.57 | 0.57 | | 11.8 |
| Hagstroem et al. (2008) | 0.70 | 0.57 | 0.80 | 3.01 | 0.00 | | 12.9 |
| Van laecke et al. (2006) | 0.36 | 0.16 | 0.63 | -1.04 | 0.30 | | 9.85 |
| Vijverberg et al.(1997-2011) | 0.68 | 0.58 | 0.77 | 3.50 | 0.00 | | 13.5 |
| Meijer et al. (2015) | 0.41 | 0.31 | 0.53 | -1.43 | 0.15 | | 13.3 |
| Hoebeke et al. (2011) | 0.41 | 0.28 | 0.57 | -1.10 | 0.28 | | 12.6 |
| Caldwell et al. (2021) | 0.22 | 0.16 | 0.31 | -5.58 | 0.00 | | 13.5 |
| | 0.49 | 0.35 | 0.63 | -0.14 | 0.89 | | |

**Fig 3.** A. Random effect model for all included RCTs with a risk ratio for continence after training. B. Random effect model including all studies for continence after training.

CI: -0.01–0.02, $Q_{model}$ = 0.07, df = 1, $P$ = 0.790). Treatment duration was not a predictor of the variance in continence rates across studies (Fig 4).

**Adherence.** A quantitative analysis of adherence was not performed as only four studies reported adherence, definitions of adherence differed, and measurements were not comparable. Boelens et al. scored nonadherence based on observations by the trainer (trying to remove the alarm, not wearing the alarm, or wearing it inappropriately) and found that three out of four patients were nonadherent [37]. Hagstroem et al. used the voiding diary as an instrument to determine nonadherence and defined being nonadherent as a long micturition interval (exceeding 3 hours during at least 3 registered days) was reported [12]. This resulted in a non-adherence rate of 33%. Halliday et al. used a mechanical measurement to define nonadherence [31]. They used a calibrated capillary tube built in the pants alarm containing a mercury level that increased if the alarm was switched on and worn, taking 100 hours to reach the end of the tube. The position was recorded 2 weeks and 1 month after treatment began. The authors found that 10% of patients wore the alarm device for less than 100 hours. Caldwell et al. defined adherence as wearing the timer watch at least 75% of the time and always or mostly

## Meta regression of treatment duration

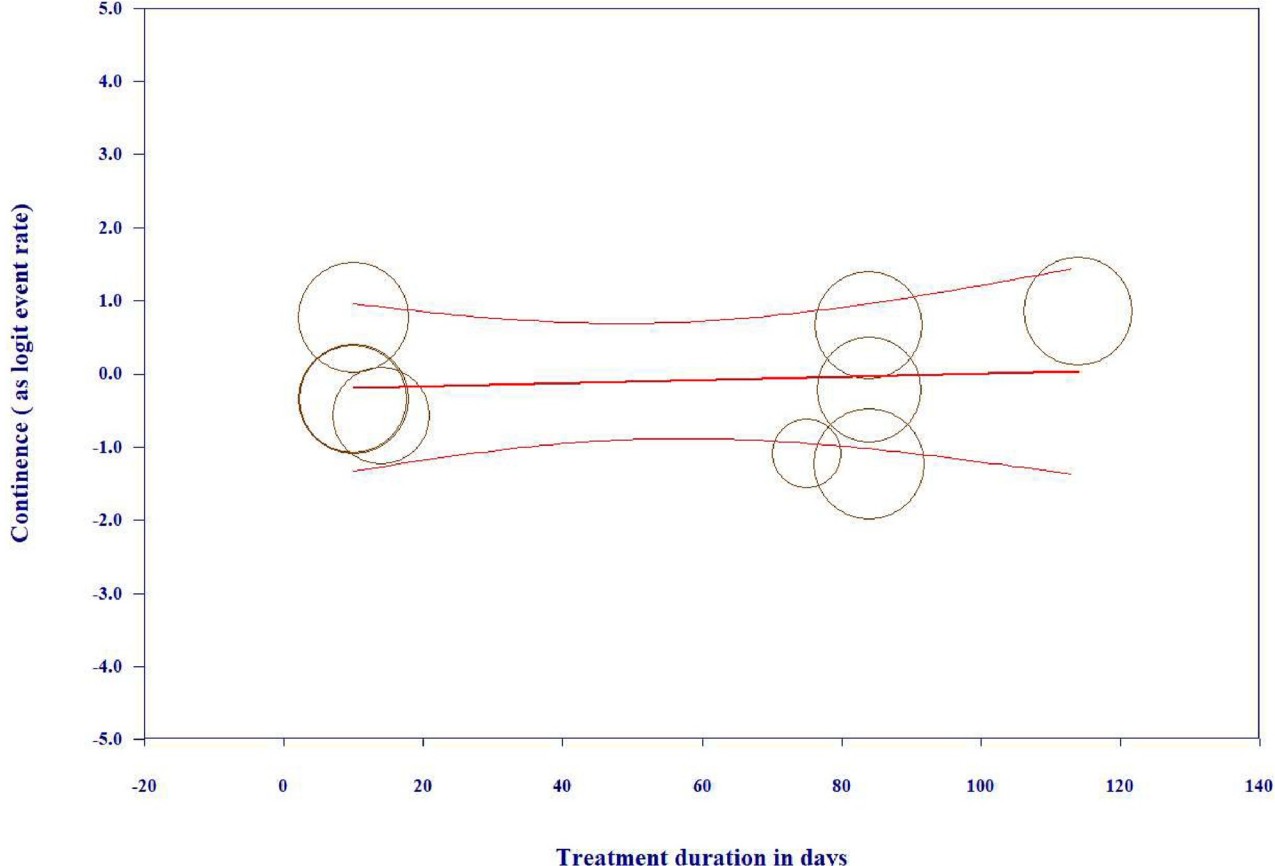

**Fig 4. Metaregression of treatment duration in days as a predictor of the variation across studies in continence rates.**

voiding at the appropriate times. The parents reported this. Caldwell et al. found rates of non-adherence of 60% and 90% for the timer watch and control groups, respectively [33].

**Publication bias/GRADE.** Inspection of the funnel plot and the trim-and-fill procedure revealed no publication bias, and the result of the Egger's test was not significant ($P = 0.867$; see Fig 5 for the funnel plot). Adjustment for missing studies did not change the effect from a Hedges $g$ of 0.49 (95% CI 0.43–0.54) to 0.49 (95% CI, 0.43–0.54). The assessed evidence of certainty (GRADE) is shown in Table 2.

## Discussion

Wearable alarm systems such as timer watches and alarm pants are additional tools in urotherapy for DUI. Although these tools are frequently used for bedwetting, we found only 10 studies that examine the effect of alarm interventions as part of a broader protocol for DUI in otherwise healthy children. Identified studies were heterogeneous in terms of quality, design, type of alarm, and treatment duration. Sample sizes were often small or underpowered, and samples were often not well defined in terms of diagnosis, concomitant treatment(s), and demographic background.

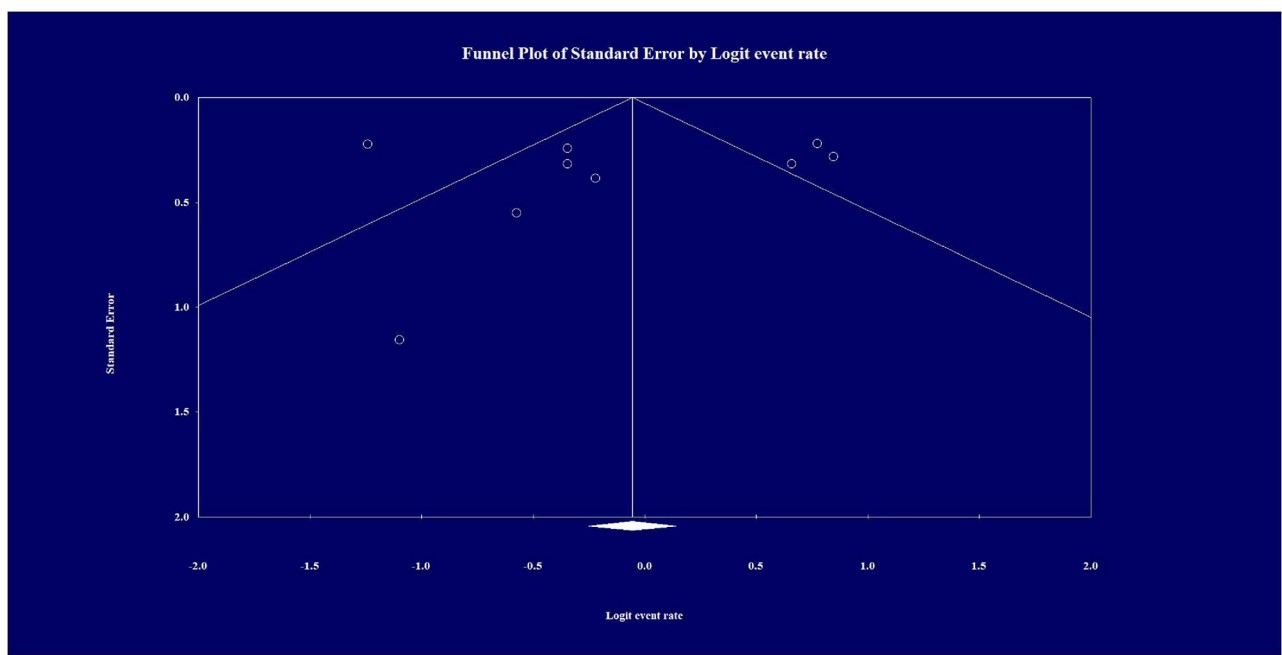

**Fig 5. Funnel plot of standard error by Hedges for trials included in the meta-analysis.** The open circles in the figure represent observed studies. The diamond represents the observed effect size. The result of the Egger's test of the intercept-indicated bias in the funnel plots was not statistically significant (*P* = 0.867).

Nonetheless, 3 RCTs were found with an RR of 17.8 for urotherapy with a timer watch versus urotherapy without and an RR of 1.2 in both other RCTs for either contingent alarm pants versus noncontingent alarm pants or timer watch versus no alarm [12,31,33]. This yielded a meta-analytic RR of 1.4 in favor of alarm interventions, though without significance (*P* = 0.257). We suspect that the large difference in RRs between the study by Hagstroem et al. and the other RCTs is caused by study design. In the RCTs by Hagstroem et al. and Caldwell et al., the intervention group (timer watch) was comparable to the control group of the study by Halliday et al. (noncontingent alarm). In addition, in the studies by both Hagstroem et al. and Caldwell et al., assessment of the outcome is likely to be at least partially influenced by knowledge of the intervention received. In the study by Hagstroem et al., the authors stated that subjective improvement occurred in 90% of the children receiving alarm treatment versus only 25% in the control group. The study by Caldwell et al. mentions that more parents from the alarm group reported that the treatment with a timer watch was helpful in reducing their

**Table 2. Assessment of evidence according to GRADE.**

| No. of studies | Design | Risk of bias | Inconsistency | Indirectness | Imprecision | Other | Dry/ alarm | Not dry/ alarm | RR/ event (95% CI) | Quality | Importance |
|---|---|---|---|---|---|---|---|---|---|---|---|
| **Quality assessment** | | | | | | | **No. of patients** | | **Effect** | **Quality** | **Importance** |
| Alarm therapy versus no alarm therapy on continence after treatment | | | | | | | | | | | |
| 3 | RCT | High | Serious | Serious | Low | None | 51 | 46 | RR 1.4 (0.8–2.6) | Very low | 2C |
| Proportional effect of alarm therapy on continence | | | | | | | | | | | |
| 6 | Observational | High | Serious | No serious indirectness | Serious | None | 136 | 277 | 48% (33–62%) | Very low | 2C |

child's wetting accidents [12,33]. A placebo effect for wearable alarms is likely to occur because the sensation of the alarm device itself increases awareness of the bladder. In the study by Halliday et al., conditions were blinded, and a noncontingent alarm was used as a sham device. In the study by Caldwell et al., a wearable timer watch with the actual alarm locked was also used for the control group. The results of their intervention group are more likely to reflect the pure effect of the alarm on continence [31,33].

We found an overall continence rate of 48% after alarm treatment (additional to urotherapy). Shaefer et al. compared a spontaneous recovery rate of functional DUI to the effect of standard urotherapy without an alarm for therapy-naïve children first diagnosed with functional DUI. They found standard urotherapy to be an effective treatment with a 54% continence rate after 1 year versus a 15% chance of spontaneous recovery within the same period [11]. This initially suggests that alarm systems are without additional value compared to urotherapy alone. However, in our meta-analysis, most studies included children with therapy refractory complaints who failed to respond to previous urotherapy. With an additional 48% continence rate in previously unsuccessfully trained children, intensifying urotherapy by including alarm interventions seems at least worth trying, especially because spontaneous recovery is not likely to occur in this group as confirmed by Hoebeke et al. [32]. Only one of the children included in their control group (without treatment) became dry. In our study, no differences appeared in continence rates between type of alarm used (i.e., timer watch or alarm pants).

## Adherence

Because urotherapy incorporates many components of CBT, motivation and adherence to treatment recommendations are crucial for success [10,39,40]. It is therefore noteworthy that only four of the nine studies included in this review described adherence. Adherence is one of the topics within the field of urotherapy that is least reported. This is probably due to several factors. First, adherence is difficult to measure as such in nonpharmacological studies. Second, and probably more important, current studies mainly focus on continence rates of different kinds of treatment modalities and less on factors determining this success, such as motivation and adherence. In this review, adherence to treatment (including alarm systems) varied between 10 and 90%. As previously mentioned, motivation in combination with a structured training program and the right support of parents and health care providers to adhere to this training program are key factors for success. Therapy-naïve and younger children might be less motivated to adhere to the treatment recommendations in general including using the alarm and writing voiding diaries, especially if training is not according to a predefined protocol and is left up to the child. In addition, DUI is likely to be present only for a short period, and quality of life is probably less affected because DUI is more common and accepted among these younger children. As children grow older and complaints persist, those still seeking medical attention are expected to be more motivated.

This hypothesis is ratified by the studies included in this review. Boelens et al. studied the use of an alarm with therapy-naïve young children (mean age of 6.5) in an unstructured training setting. Treatment duration, contact moments and number of days the alarm was worn were not protocolized. Nonadherence was high at 75% [37]. Halliday et al. found the lowest levels of nonadherence (10%) in therapy-resistant children of an older age (mean age 8.2) throughout a predefined training program [31].

## Treatment duration and relapse

The amount of time in which alarm interventions were used as part of urotherapy varied between 10 days and a maximum of 146 days [32,34]. Studies in which the alarm intervention

was used for a short period obtained similar effects as those with longer periods of use. Within CBT, the number of sessions positively predicts the outcome directly after treatment but not during follow-up [41]. Because urotherapy incorporates many aspects of CBT, intense training with an alarm throughout a short training period might be as beneficial as less intense guidance during a longer training period. Boelens et al. mention that the results of the first week of training predict the success [37]. Van Laecke used the alarm for only two weeks but obtained similar effects as Hagstroem et al. in his RCT, where some children were still dependent on the alarm 7 months after treatment [12,16].

In this study, relapse after complete response during a follow-up of 6 months or more varied between 0 and 43% and seemed more common in the studies with shorter treatment periods (10–14 days of inpatient training) than those studies including the alarm intervention for 12 consecutive weeks [12,16,31,33,34,36]. However, studies with higher relapse rates all included therapy-refractory children who were still wet after ambulant training. Inpatient training was considered a last resource option. This explains the higher relapse rates in these children compared to those who trained at home. Adherence, support throughout training, dependency prevention, and relapse after successful treatment are all factors that play a role in determining the optimal duration for alarm interventions used. Based on the above-mentioned results, using alarm devices for prolonged periods may increase the risk of children becoming dependent, and shorter periods also seem effective. However, the exact duration needs to be determined.

## Strength and limitations

In a Cochrane review studying all conservative interventions including alarm systems for treating functional DUI in children, it remained uncertain whether the addition of timer watches was more effective in reducing incontinence compared to urotherapy alone [42]. However, this conclusion was only based on one study. To our knowledge, this meta-analysis is the first study to include both RCTs and observational studies and evaluate individual aspects of urotherapy such as alarm interventions more specifically.

Limitations of this study include the small number of studies examined based on the limited number of available studies addressing alarm interventions for DUI in children in addition to the low methodological quality of the existing literature. Differences in included children, type of alarm used, treatment protocols, and definitions of success make it difficult to offer reliable conclusions. In our review, the high likelihood of confounding bias might have influenced outcomes. Confounding bias is likely to occur in any trial studying the effects of a particular intervention (like alarm intervention) as part of a broader treatment. RCTs including a control group (without an alarm) and a group with a sham device in addition to the actual intervention group are recommended to study the real effect of alarm interventions added to urotherapy. Future research should focus on studies of a high methodological quality including a control group and sham device with the use of clearly defined inclusion criteria (therapy-naïve versus therapy-refractory children, exact duration of treatment/use of device, definition outcome according to ICCS criteria, and definitions for adherence).

## Conclusion

Evidence about the additional value of alarm interventions in urotherapy is sparse and heterogeneous but in slight favor of the use of wearable alarm systems for children with functional DUI. Effectiveness ranges between 33 and 62%. In future research, it is recommended to imbed both adherence and treatment duration as study outcomes to obtain a better idea of optimal treatment protocol and sustainability of treatment effects.

## Supporting information

**S1 Checklist. PRISMA 2009 checklist.**
(PDF)

**S1 Appendix. Full electronic search.**
(DOCX)

## Author Contributions

**Conceptualization:** Liesbeth L. de Wall, Antje J. Nieuwhof-Leppink, Renske Schappin.

**Data curation:** Liesbeth L. de Wall, Antje J. Nieuwhof-Leppink.

**Formal analysis:** Liesbeth L. de Wall.

**Investigation:** Liesbeth L. de Wall.

**Methodology:** Liesbeth L. de Wall, Renske Schappin.

**Project administration:** Liesbeth L. de Wall.

**Software:** Liesbeth L. de Wall.

**Supervision:** Antje J. Nieuwhof-Leppink, Renske Schappin.

**Validation:** Antje J. Nieuwhof-Leppink, Renske Schappin.

**Visualization:** Liesbeth L. de Wall.

**Writing – original draft:** Liesbeth L. de Wall.

**Writing – review & editing:** Liesbeth L. de Wall, Antje J. Nieuwhof-Leppink, Renske
  Schappin.

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
