## [Decision Letter · Decision Letter 0]

12 Jul 2022

PONE-D-21-35570Alarm-assisted urotherapy for daytime urinary incontinence in children. A meta-analysis.PLOS ONE

Dear Dr. Wall,

Thank you for submitting your manuscript to PLOS ONE; I sincerely apologise for the unusually delayed review timeframe. Your manuscript has been assessed by two reviewers, whose comments are appended below. After careful consideration, we feel that it has merit but does not fully meet PLOS ONE’s publication criteria as it currently stands. Therefore, we invite you to submit a revised version of the manuscript that addresses the points raised during the review process. In particular, please ensure that the statistical analysis and data presentation is revised per the recommendations of reviewer 2.

We look forward to receiving your revised manuscript.

Kind regards,

Emily Chenette

Editor in Chief

PLOS ONE

Journal Requirements:

2. Thank you for stating the following financial disclosure: "The funders had no role in study design, data collection and analysis, decision to publish, or preparation of the manuscript." 

Reviewers' comments:

Reviewer's Responses to Questions

**Comments to the Author**

1. Is the manuscript technically sound, and do the data support the conclusions?

Reviewer #1: Yes

Reviewer #2: Yes

2. Has the statistical analysis been performed appropriately and rigorously? 

Reviewer #1: Yes

Reviewer #2: Yes

3. Have the authors made all data underlying the findings in their manuscript fully available?

Reviewer #1: Yes

Reviewer #2: Yes

4. Is the manuscript presented in an intelligible fashion and written in standard English?

Reviewer #1: Yes

Reviewer #2: Yes

5. Review Comments to the Author

Reviewer #1: 1. Please make sure that "data" are used as plural throughout

2. On line 113, “trials” is misspelled twice

3. Please comment further based on your thorough review on what is needed from the literature on the topic in future studies

Reviewer #2: The study aims to systematically evaluate, summarize, review, and analyze existing evidence about the effect of wearable alarm systems in urotherapy for children with functional DUI (daytime urinary incontinence).

This is an interesting study, well-analyzed and presented. However, the manuscript can be further improved based on the following comments.

Data synthesis and statistical analysis

Line 179-185, the use of fixed effect and random effect is not clear and the sentence requires revision.

The software that was used in performing the analysis is to be stated.

Results

Table 1, the title is too short. In the treatment duration days column, some indicated protocol and few without protocol. For Intervention and control condition column, for those that indicated a single group apart from those studies which had intervention and control, more information is to be provided. Dash(-) to be denoted in the table footnote.

The subtitles in Table 1 could be reworded. For example, state authors (year), country, study design, mean age (sd, range).

For Median 114 [CI 64-146], the level of CI is to be stated.

For the nonadherence 10%, how many were from intervention and control to be stated. More information could be provided in the table to provide more information to readers.

Table 2, CI 95 to be rewritten as 95%CI. Grade to GRADE.

Study outcomes

Line 256, the use of OR for RCTs is inappropriate,

Line 266, what are outliners?

Secondary outcomes

Line 279, the representation of variance component to be clearly stated.

Treatment duration

Line 299, the R^2 value to be stated.

At least one decimal point for the percentage figure to be provided.

P or italicize P to be standardized throughout.

Figure 1, the exclusion of n=19 to be separated and denoted with an asterisk for the individual items that sum up to n=19

Figure 3B, event rate to be denoted in the table footnote.

Figure 3A, 3B, Figure 5, the decimal points for the data were converted to commas in the PDF format.

The citation of references in the text and list of references to follow journal format.

The manuscript requires some English editing.

6. PLOS authors have the option to publish the peer review history of their article (what does this mean?). If published, this will include your full peer review and any attached files.

Reviewer #1: No

Reviewer #2: No

---

## [Author Response · Author response to Decision Letter 0]

8 Sep 2022

Reviewer 1 

1. Please make sure that “data” are used as plural throughout. 

Thank you for the remark, we adjusted it throughout the manuscript.

2. On line 113, “trials” is misspelled twice.

Thank you for the comment. A native speaker edited the whole manuscript and the manuscript was adjusted accordingly. This included the misspelled “trials”. 

3. Please comment further based on your thorough review on what is needed from the literature on the topic in future studies.

To clarify future needs on this topic the paragraph strength and limitations was extended, see also line 460-463;

“Future research should focus on high methodological quality research including a control group and sham device with use of clear defined inclusion criteria (therapy naïve versus therapy refractory children, exact duration of treatment/use of device, definition outcome according to ICCS criteria and definitions for adherence).”

Reviewer 2

The study aims to systematically evaluate, summarize, review and analyze existing evidence about the effect of wearable alarm systems in urotherapy for children with functional DUI (daytime urinary incontinence). This is an interesting study, well-analyzed and presented. However, the manuscript can be further improved based on the following comments:

1. Data synthesis and statistical analysis

Line 179-185, the use of fixed and random effects Is not clear and the sentence requires revision.

Thank you for the remark. 

In a fixed-effect model it is assumed that all studies in the meta-analysis share a common (true) effect size. Or in others words, all factors that could influence the effect size is the same for all the individual studies. This is often implausible as the studies are not completely identical in all aspects. Therefore a random effect model is preferable. 

Initially with only two RCT’s the estimate of the between studies variance will have poor precision and cannot be applied appropriately. A fixed effect model seemed the second best option to at least get an idea of the summarize effect size of both RTC’s. 

However, due to the long time between initial submission and review (November 2011-July 2022) the literature was not up to date anymore. To overcome this, we decided to repeat our complete search. This led to an extra published RCT in the meantime and gave us the opportunity to do a preferred random-effect model including three instead of two RCT’s. 

We additionally adjusted the section: “Data synthesis and statistical analysis” accordingly by mentioning the random effect model to estimate overall effect size of alarm intervention added to urotherapy.

2. The software that was used in performing the analysis is to be stated.

Thank you for the comment. In the revised manuscript the software used for data analysis is mentioned in line 190 (including its reference).We tried to rephrase the sentence as we understand CMA might not be a familiar software program.

“ The software used to analyze the data was Comprehensive Meta-Analysis V3 . [30]”

3. Table 1, the title is too short

We adjusted the title to: “Study characteristics of all studies included in this systematic review.”

4. In the “treatment duration, days” column, some indicated protocol and few without protocol. Dash(-) to be noted in the footline

Thank you for the comment. We understand this might be confusing. If a protocol was used in the included study, the actual days of alarm use was mentioned. In some studies no specific protocol was used and the alarm was worn for a different period of time in each individual child. In that case either the median of mean (upon available in the article) was mentioned. A footnote is added to clarify this. The meaning of (-) is also included in the footnote.

5. For the “Intervention and control condition column”, for those that indicated a single group apart from the studies which had intervention and control more information is to be provided. 

Thank you for the comment. We adjusted it in the Table and renamed it into “ Type of alarm” for better clarification. This was further subclassified in case of more than one group was present.

6. The subtitles in Table 1 could be reworded. For example, state authors (year), country, study design, mean age (sd, range). For Median 114 [CI 64-146], the level of CI is to be stated.

Thank you for the comment. We reworded the content- including subtitles and footnote- to clarify its content. The level of CI is stated.

7. For the nonadherence 10%, how many were from intervention and control to be stated. 

Unfortunately the authors of this specific study did not further subclassified the 10% non-adherence into those who had used the contingent or non-contingent alarm. Due to its publication date in 1987, this information could not be retrieved in another way. This is added to the table/ footnote. The table provides all available information retrieved from the included study. However, due to the low methodologic quality of some of the studies some information is simply not available. 

8. Table 2, CI 95 to be rewritten as 95%CI. Grade to GRADE.

Thank you for the comment. This was adjusted throughout the manuscript.

9. Line 256, the use of OR for RCTs is inappropriate.

Thank you for the comment. We adjusted it to risk ratio (RR).

10. Line 266, what are outliners?

Thank you for the comment, this should be outliers. Outliers are studies with potential impact on the overall summary effect like the study of Boelens et al. The continence rate after treatment in this study is relatively low compared to the other studies included in this Meta-Analysis. By performing a sensitivity analyses excluding this study in particular we could determine how robust our findings were and how much the summary effect would change. We rephrased (line 279-281) the sentence into :

“Subsequent sensitivity analyses, which removed the study by Boelens et al. because the relatively low continence rate in this study might potentially influence summary effect, revealed a summary effect comparable to the main analyses, namely a 48% continence rate (95% CI: 35–64%).”

11. Line 279, the representation of variance component to be clearly stated.

Thank you for the comment. We adjusted this in the sections: “ Data analysis and statistical analysis.” Line 198:

“To investigate if relapse, treatment duration, or adherence were relevant predictors that could account for the variance component Tau2 in the overall event rate, a meta-regression analysis was performed assuming a random effect model, using a Z distribution.”

It was further adjusted in the section “Type of Alarm”, line 289:

“A subgroup analysis revealed 49% continence (95% CI:32-67%) after treatment if a pants alarm was used versus 45% continence (95% CI: 23-68%) when a timer watch was used, Qbetween=0.10, df =1, P=0.751, Tau2 =0.63.”

13. Line 299, the R^2 value to be stated.

Thank you for the comment. We adjusted this in the sections: “ Data analysis and statistical analysis.” Line 199:

“The regression coefficient R2 was used as a measure of the proportion of the observed variance that was not due to sampling error.”

It was further adjusted in the section “Treatment duration”, line 311:

The overall proportion of variance explained by treatment duration was 0% (R2 = 0.00, 95% CI: -0.01-0.02, Qmodel =0.07, df =1, P=0.790).

14. At least one decimal point for the percentage figure to be provided.

Thank you for the comment. We would like to ask the reviewer for clarification which specific Figure needs to be adjusted as to our knowledge there is no figure with percentages.

15. P or italicize P to be standardized throughout.

Thank you for the comment. This was adjusted throughout the manuscript.

16. Figure 1, the exclusion of n=19 to be separated and denoted with an asterisk for the individual items that sum up to n=19.

We adjusted the figure to clarify the individual items of the total of 19.

Figure 3B, event rate to be denoted in the table footnote.

The footnote/ title was adjusted. 

Figure 3A, 3B, Figure 5, the decimal points for the data were converted to commas in the PDF format.

Thank you for the comment, the figures are adjusted.

17. The citation of references in the text and list of references to follow journal format.

Thank you for the comment. The reference list and citations are adjusted to the journal format.

In addition we updated the reference list with relevant current literature.

Two references were retracted: 

Hellstrom AL, Hanson E, Hansson S, Hjalmas K, Jodal U. Micturition habits and incontinence in 7-year-old Swedish school entrants. Eur J Pediatr. 1990;149(6):434-7.

Swithinbank LV, Heron J, von Gontard A, Abrams P. The natural history of daytime urinary incontinence in children: a large British cohort. Acta Paediatr. 2010;99(7):1031-6.

And replaced by:

Linde JM, Nijman RJM, Trzpis M, Broens PMA. Prevalence of urinary incontinence and other lower urinary tract symptoms in children in the Netherlands. J Pediatr Urol. 2019;15(2):164 e1- e7. Epub 20181108. doi: 10.1016/j.jpurol.2018.10.027. PubMed PMID: 30583907.

Xing D, Wang YH, Wen YB, Li Q, Feng JJ, Wu JW, et al. Prevalence and risk factors of overactive bladder in Chinese children: A population-based study. Neurourol Urodyn. 2020;39(2):688-94. Epub 20 

 In addition a new study was added to the systematic review:

 Caldwell PHY, Kerr M, Hamilton S, Teixeira-Pinto A, Craig JC. An Alarm Watch for Daytime Urinary 

Incontinence: A Randomized Controlled Trial. Pediatrics. 2022;149(1). doi: 10.1542/peds.2021-053863. PubMed PMID: 34907443.

 

16. The manuscript requires some English editing.

Thank you for the comment. The revised manuscript has been reviewed by a native speaker and adjusted accordingly.

---

## [Decision Letter · Decision Letter 1]

27 Sep 2022

Alarm-assisted urotherapy for daytime urinary incontinence in children. A Meta-Analysis.

PONE-D-21-35570R1

Dear,

We’re pleased to inform you that your manuscript has been judged scientifically suitable for publication and will be formally accepted for publication once it meets all outstanding technical requirements.

Kind regards,

Muhammad Shahzad Aslam, Ph.D.,M.Phil., Pharm-D

Academic Editor

PLOS ONE

Additional Editor Comments (optional):

Reviewers' comments:

Reviewer's Responses to Questions

**Comments to the Author**

1. If the authors have adequately addressed your comments raised in a previous round of review and you feel that this manuscript is now acceptable for publication, you may indicate that here to bypass the “Comments to the Author” section, enter your conflict of interest statement in the “Confidential to Editor” section, and submit your "Accept" recommendation.

Reviewer #2: All comments have been addressed

2. Is the manuscript technically sound, and do the data support the conclusions?

Reviewer #2: Yes

3. Has the statistical analysis been performed appropriately and rigorously? 

Reviewer #2: Yes

4. Have the authors made all data underlying the findings in their manuscript fully available?

Reviewer #2: Yes

5. Is the manuscript presented in an intelligible fashion and written in standard English?

Reviewer #2: Yes

6. Review Comments to the Author

Reviewer #2: The percentage figures in Table 1 are without decimal points. Please check if the actual figures are with or without decimal points.

7. PLOS authors have the option to publish the peer review history of their article (what does this mean?). If published, this will include your full peer review and any attached files.

Reviewer #2: No

---

## [Editor Report · Acceptance letter]

3 Oct 2022

PONE-D-21-35570R1 

Alarm-assisted urotherapy for daytime urinary incontinence in children: A meta-analysis. 

Dear Dr. Wall:

I'm pleased to inform you that your manuscript has been deemed suitable for publication in PLOS ONE. Congratulations! Your manuscript is now with our production department. 

Kind regards, 

on behalf of

Dr. Muhammad Shahzad Aslam 

Academic Editor

PLOS ONE